# A Novel Walking Activity Recognition Model for Rotation Time Series Collected by a Wearable Sensor in a Free-Living Environment

**DOI:** 10.3390/s22093555

**Published:** 2022-05-07

**Authors:** Raphaël Brard, Lise Bellanger, Laurent Chevreuil, Fanny Doistau, Pierre Drouin, Aymeric Stamm 

**Affiliations:** 1Department of Mathematics Jean Leray, UMR CNRS 6629, Nantes University, 44322 Nantes, France; rbrard@umanit.fr (R.B.); lise.bellanger@univ-nantes.fr (L.B.); pdrouin@umanit.fr (P.D.); 2UmanIT, 13 Place Sophie Trébuchet, 44000 Nantes, France; lchevreuil@umanit.fr (L.C.); fdoistau@umanit.fr (F.D.)

**Keywords:** machine learning, human activity recognition, walk detection, IMU, unit quaternion time series, time series segmentation

## Abstract

Solutions to assess walking deficiencies are widespread and largely used in healthcare. Wearable sensors are particularly appealing, as they offer the possibility to monitor gait in everyday life, outside a facility in which the context of evaluation biases the measure. While some wearable sensors are powerful enough to integrate complex walking activity recognition models, non-invasive lightweight sensors do not always have the computing or memory capacity to run them. In this paper, we propose a walking activity recognition model that offers a viable solution to this problem for any wearable sensors that measure rotational motion of body parts. Specifically, the model was trained and tuned using data collected by a motion sensor in the form of a unit quaternion time series recording the hip rotation over time. This time series was then transformed into a real-valued time series of geodesic distances between consecutive quaternions. Moving average and moving standard deviation versions of this time series were fed to standard machine learning classification algorithms. To compare the different models, we used metrics to assess classification performance (precision and accuracy) while maintaining the detection prevalence at the level of the prevalence of walking activities in the data, as well as metrics to assess change point detection capability and computation time. Our results suggest that the walking activity recognition model with a decision tree classifier yields the best compromise in terms of precision and computation time. The sensor that was used had purposely low computing and memory capacity so that reported performances can be thought of as the lower bounds of what can be achieved. Walking activity recognition is performed online, i.e., on-the-fly, which further extends the range of applicability of our model to sensors with very low memory capacity.

## 1. Introduction

Loss of autonomy greatly impacts quality of life in general. The ability to walk has been reported to be the most important skill for people to feel autonomous because it is fundamental to most of our daily routines [1]. It is therefore critical to provide solutions that can help to monitor gait. Human gait analysis is defined as the systematic study of human motion. This is accomplished by measuring a number of parameters that characterize the movements of the human body [1,2,3,4]. Humans walk in a mostly periodic way composed of steps known as *gait cycles*. These gait cycles are in turn divided into two parts: a *stance phase*, accounting for about 60% of the gait cycle, and a subsequent swing phase, accounting for the remaining 40% of the cycle [1]. We distinguish *kinetics* from *kinematics* data. The former collect information about strength and muscle activity during motion while the latter collect information about the velocity and acceleration of body segments, joint angles between body segments, etc. Collecting these data usually requires heavy equipment and a dedicated room for making the measurements  [3,4,5].

The advent and development of wearable sensors over the last decade offers a natural alternative as a response to this criticism [2,6,7,8,8]. In effect, they are lightweight and a lot less expensive than equipping an entire room with cameras and sensors to record every move [3,4,5]. Equipping people with enough tiny motion sensors on body parts such as the foot [2,9] or the waist [10,11], and post-processing the data using appropriate mathematical methods and models, is very effective for reaching results similar to those obtained from dedicated gait monitoring rooms [3,4,5,12]. In addition, since motion sensors are lightweight and discrete, they can be worn in daily life without constraint and can therefore provide the analyst with continuous-time gait information for a better more objective and bias-free monitoring. This has been done for monitoring leg injuries or post-surgical remission to propose targeted re-education strategies [13,14] or for detecting loss of independence among the elderly in everyday life [15,16,17].

The main limitation of wearable sensors for monitoring gait in daily life situations is that the collected data does not contain only phases during which the subject is actually walking. To circumvent this issue, most gait monitoring solutions based on wearable sensors either collect data in laboratory [2,18] or in a semi free-living environment with instructions to keep walking during recording [7,8]. However, everyday life gait-monitoring solutions are key for studying pathologies such as Parkinson’s disease in which gait is not impacted equally during all times of day. This type of data preprocessing is known in the literature as human activity recognition (HAR) and can be decomposed into the following four steps [19]:Data acquisition and cleaning;Data transformation: this step aims at creating novel relevant variables which shall help subsequent steps;Data segmentation: this step involves change point detection methods to identify time points along the time series that correspond to a change in activity and output the corresponding segments.Segment classification: this step aims at merging the previously determined segments into groups that correspond to distinct activities.

In this paper, we aim at designing a novel walking activity recognition model for separating walking activity phases from non-walking activity phases using data collected by wearable sensors. We designed our model by putting ourselves in the worst case scenario on several aspects: (i) we used hip rotation data which might be less sensitive to gait than body parts closer to the feet [20] but presents the advantage of being so discrete and non-invasive that the patient forgets (s)he is wearing it which mitigates the observation bias, (ii) we used a wearable sensor with low computing power and low memory and (iii) we targeted real-time (on-the-fly) walking activity detection which is especially important for low memory devices so that we only store the relevant data. We hypothesize that designing a walking activity recognition model from data that fulfill these conditions will provide the most general model for detecting walking activity with expected improved performances when more sophisticated gait-measuring devices are instead used.

When monitoring gait kinematics, the collected data naturally comes in the form of three-dimensional rotations. Traditional HAR models can be applied seamlessly on time series evaluating in Euclidean spaces [19,21,22]. Rotation time series, however, evaluate on the 3-sphere, which is a non-Euclidean manifold. Therefore, while taking inspiration from the HAR pipeline, we take on a slightly different approach to accommodate the non-Euclidean nature of our data. Specifically, we make two main contributions:In Step 2, we convert unit quaternion time series into real-valued time series designed to facilitate subsequent segmentation and classification;We combine Steps 3 and 4 into a single step. We indeed segment and classify simultaneously by training a classifier from annotated data.

Figure 1 summarizes the proposed modeling strategy. To choose the *best* classifier, we rigorously compared the most common supervised classification algorithms used in the HAR literature [22,23], namely, decision trees [24,25,26], support vector machine [10,19,24,25,26,27,28], k-nearest neighbors [24,25,27,28] and logistic regression [28,29]. The notion of *best* is intended in terms of both classification performances and short computation time because the proposed model is to be implemented on the sensor chip itself.

In summary, we present in detail an efficient and complete model to perform the detection of walking phases from rotation data recorded in real life settings by a wearable sensor. Section 2 describes the proposed pipeline in depth. Specifically, we explain (i) how the annotated data were designed (*data acquisition step*), (ii) which real-valued time series we extracted from the rotation time series and why we think these new time series are particularly relevant to the task of detecting walking phases (*data transformation step*) and (iii) how the different classification algorithms work and which metrics we used to tune and compare them (*combined segmentation and classification step*). Section 3 is dedicated to describing how we tuned the algorithms from the training set and to exposing the performance of the different tuned classifiers on the test set. Finally, Section 4 provides a discussion of the results along with some recommendations for choosing an optimal walking phase detection algorithm for integration on the sensor chip. All the codes were developed in the R language for statistical computing [30].

## 2. Proposed Walking Activity Recognition Model

### 2.1. Data Acquisition

We use a MetaMotionR motion sensor [31] to measure hip rotation over time. Figure 2 shows pictures of this sensor alongside with an individual, which demonstrates how non-invasive it is. The MetaMotionR (MMR) sensor contains an *Inertial Measurement Unit (IMU)*. The IMU is made of three orthogonal accelerometers, three orthogonal gyroscopes and three orthogonal magnetometers, mutually aligned to each other. The data collected by these nine individual sensors are combined by a sensor fusion algorithm [32] implemented on the chip itself. The MMR sensor has four fusion modes: (i) NDoF which calculates absolute orientation from accelerometer, gyroscope, and magnetometer data, (ii) IMUPlus which calculates relative orientation in space from accelerometer and gyroscope data, (iii) Compass which determines geographic direction from the Earth’s magnetic field and (iv) M4G which is similar to IMUPlus except rotation is detected with the magnetometer. For the data acquisition, we chose the NDoF fusion mode to obtain the absolute orientation vector of the device in the form of a **unit quaternion** and we used the Android application *MetaBase App* developed by *Mbientlab* to control the device via *Bluetooth*. We set the options to *Quaternion mode* with a sample rate of 100 Hz and to *log mode*, such that the data are stored in the device memory before its exportation to the smartphone via *Bluetooth*. The settings of the device used for the acquisition are described in Table 1.

A quaternion is a mathematical object suitable for representing three-dimensional rotations. It is defined as follows:

**Definition** **1**(Quaternion). *A quaternion q is a four-dimensional vector that can be written as:*
(1)q=w+xi+yj+zk,
*where w, x, y and z are real numbers and i, j and k are pure complex numbers which satisfy $i^{2}=j^{2}=k^{2}=ijk=-1$. We denote by H the space of quaternions. More details about this space can be found in Appendix A.*

In particular, the sensor computes a **unit quaternion** every 10 ms, which represents the rotation that brings the hip from its orientation at the beginning of the recording to its current orientation. The statistical unit in our data set is thus a **unit quaternion time series** (uQTS) which can be formally defined as:

**Definition** **2**(Unit Quaternion Time Series). *Let $t={\left(t_{1},\cdots ,t_{P}\right)}^{⊤}$ be a grid of P time points separated by a constant step of Δt=10 ms. We define a*
***unit quaternion time series***
*(uQTS) on this grid as:*
(2)$$q\left(t\right)={\left(q\left(t_{1}\right),\cdots ,q\left(t_{P}\right)\right)}^{⊤},$$
*where $q\left(t_{\ell }\right)=w\left(t_{\ell }\right)+x\left(t_{\ell }\right)i+y\left(t_{\ell }\right)j+z\left(t_{\ell }\right)k\in H$ with $∥q\left(t_{\ell }\right){∥}^{2}=w{\left(t_{\ell }\right)}^{2}+x{\left(t_{\ell }\right)}^{2}+y{\left(t_{\ell }\right)}^{2}+z{\left(t_{\ell }\right)}^{2}=1$ for all ℓ=1,⋯,P.*

To the best of our knowledge, there has not been any experimental design in the literature for this type of data. Already published works usually focus either on acceleration, gyroscopic or magnetometer data [11,33], or on GPS data [9,34]. The purpose of this work is to identify walking activity phases in a data set measuring daily life activities. These walking phases are intended to be fed into subsequent algorithms to achieve an objective and quantitative charaterization of the gait in the form of spatio-temporal parameters or kinematics parameters. We chose the quaternion representation for rotation data for the following three reasons:Estimating the orientation of a device by integration of the angular velocity measured by the gyroscopes is known to be biased by the presence of a drift that increases with the acquisition time. This leads to unreliable measures [35], especially in the context of recording daily life activities over long periods of time. The use of sensor fusion algorithms is a popular strategy to overcome this issue [20].Unit quaternions are very convenient for representing 3D rotations and orientations, as they are less computationally demanding than *rotation matrices* or *Euler angles* and they do not suffer from the *gimbal lock* [36].Some gait analysis methods rely on representing the orientation of body segments with unit quaternions to determine the gait kinematic or spatio-temporal parameters [37,38,39,40]. These methods can therefore be directly applied on the walking activity phases identified with the proposed algorithm to quantify several aspects of gait.

A proper HAR method is therefore needed to identify walking activity phases from quaternion time series data. As stated before, current HAR methods are not suitable for this data type. Nevertheless, all these works aim at performing the same task of tuning and comparing machine learning algorithms for solving HAR problems. As such, they share a number of commonalities in their experimental design which we can take inspiration from. For instance, in the works of Ortiz [33], Anguita et al. [41], Ortiz [42], Kwapisz et al. [11], Garcia-Gonzalez et al. [34] and Beaufils et al. [9], we can grasp two key experimental setup features: (i) the type of activities are kept relatively simple among all types of activities that an individual can subject themselves to in a free living environment and (ii) it is important to mark a pause in-between two activities when building the training set because it facilitates time point labelling in the different activities.

Since the motivation of our solution is to monitor walking deficiencies, we therefore decided to limit the type of walking activities to the following list: straight line, curved line, walk up and down the stairs and sit on/stand up from a chair. We did not include running or jumping activities. We included a pause of 3 s at each change of activity. The final design of the walking path that we adopted is sketched in Figure 3.

The final data set is composed of 28 different trials of this walking path recorded on 3 different healthy volunteers (2 men and 1 woman). Every participant walked this path 10 times on average, with randomly assigned times of day (night excluded) to avoid any bias induced by daily routine. Table 2 summarises the information pertaining to the acquired data.

The total length of recording was about 40 min. One can notice from Table 2 that the data set is well balanced since each participant made approximately the same number of trials. Moreover, each run was about 90 s long, regardless of the participant. Overall, all subjects are equally represented in the data set. Figure 4 shows an example of uQTS as recorded by the motion sensor when an individual performed the walking path described in Figure 3.

### 2.2. Feature Space

Unit quaternions belong to a non-Euclidean manifold and, as such, most machine learning algorithms do not handle such data as predictors. As a result, it can be useful to transform uQTS into real-valued time series because that will provide access to all existing machine learning classifiers. In addition, we strongly believe that using less data in terms of *quantity of variables* but more data in terms of *informativeness* is key to estimate more interpretable yet still competitive models.

Given the uQTS q(t), the quaternion $q_{\ell }:=q\left(t_{\ell }\right)$ measured at time $t_{\ell }$ represents the rotation that brings the hip from its orientation at the beginning of the recording to its current orientation. Consequently, we can define the following time series:

**Definition** **3**(Quaternion Finite Difference Time Series). *Let $q\left(t\right)={\left(q_{1},\cdots ,q_{P}\right)}^{⊤}:={\left(q\left(t_{1}\right),\cdots ,q\left(t_{P}\right)\right)}^{⊤}$ be a uQTS defined on grid $t=(t_{1},\cdots ,t_{P})$. We define the*
***quaternion finite difference time series***
*(QFDTS) of a uQTS as:*
(3)$$\Delta q\left(t\right)={\left(q_{1}^{-1}q_{2},\cdots ,q_{P-1}^{-1}q_{P}\right)}^{⊤},$$
*where the inverse and (Hamiltonian) product for quaternions are defined in Appendix A.*

Observe that Δq(t) is in fact defined on a grid of size P−1. Each element of the QFDTS is again a unit quaternion which represents the rotation that brings the hip from its orientation at the previous time point to its orientation at the current time point. This is a more practical uQTS for the task of detecting changes in activity type because one can make the reasonable assumption that the hip rotation in an interval of 10 ms should be negligible during a given type of activity, but significantly larger at the transition between two types of activity. Following this line of reasoning, it makes sense to subsequently compute the real-valued time series of the norm of quaternions in Δq(t), which is nothing but the time series of the *geodesic distances* between two consecutive quaternions (see Appendix A). This time series will in fact represent the angle in radian between two consecutive quaternions.

**Definition** **4**(Quaternion Distance Time Series). *We define the real-valued time series of geodesic distances between consecutive quaternions in a uQTS q(t), coined*
***quaternion distance time series***
*(QDTS) of the uQTS, as:*
(4)d(t)=(d1,⋯,dP−1)⊤=∥q1−1q2∥,⋯,∥qP−1−1qP∥⊤=2arccosRe(q1−1q2),⋯,arccosRe(qP−1−1qP)⊤

Figure 5 shows an example of QDTS computed from a uQTS.

We can observe that raw QDTS are, in general, very noisy, which is as expected since, by definition, they rely on a finite difference scheme typically used to numerically approach derivatives. We therefore used a *sliding window* to produce a smoothed version of the QDTS. Keeping in mind that the solution should later be implementable on the sensor chip to label time points as walking or non-walking activity *in a streamlined fashion*, we use a *left hand-side sliding window* instead of a centered one because future time points will, by definition, not be available for smoothing a given time point. We therefore introduce the following

**Definition** **5**(Local Mean Quaternion Distance Time Series). *Let $d\left(t\right)={\left(d_{1},\cdots ,d_{P}\right)}^{⊤}$ be a QDTS as introduced in Definition 4. Given a window size of h, we define the*
***local mean quaternion distance time series***
*(LM-QDTS) as:*
(5)$$d^{\left(h\right)}\left(t\right)={\left(d^{\left(h\right)}\left(t_{1}\right),\cdots ,d^{\left(h\right)}\left(t_{P}\right)\right)}^{⊤}={\left(d_{1}^{\left(h\right)},\cdots ,d_{P}^{\left(h\right)}\right)}^{⊤},$$
*where $d_{\ell }^{\left(h\right)}$ is the circular mean of the angles $d\left(t_{\max(\ell -h,1)}\right),\cdots ,d\left(t_{\ell }\right)$, which reads [43]:*
$$d_{\ell }^{\left(h\right)}=atan2\left(\sum_{m=\max(\ell -h,1)}^{\ell }\sin d\left(t_{m}\right),\sum_{m=\max(\ell -h,1)}^{\ell }\cos d\left(t_{m}\right)\right).$$

Observe that, consistently, if the window size is h=0, then $d^{\left(0\right)}=d$ as expected. Figure 6 shows an example of LM-QDTS computed from a uQTS.

The LM-QDTS is naturally a good candidate to use as a predictor to label time points in different types of activities. Nevertheless, if an individual switches activities too quickly, the LM-QDTS might miss the transitions. We therefore introduce a second real-valued time series that computes a local circular standard deviation on the LM-QDTS.

**Definition** **6**(Local Standard Deviation Quaternion Distance Time Series). *Given a window size of h, let $d^{\left(h\right)}\left(t\right)={\left(d_{1}^{\left(h\right)},\cdots ,d_{P}^{\left(h\right)}\right)}^{⊤}$ be a LM-QDTS as introduced in Definition 5. We define the*
***local standard deviation quaternion distance time series***
*(LSD-QDTS) as:*
(6)$$\sigma^{\left(h\right)}\left(t\right)={\left(\sigma^{\left(h\right)}\left(t_{1}\right),\cdots ,\sigma^{\left(h\right)}\left(t_{P}\right)\right)}^{⊤}={\left(\sigma_{1}^{\left(h\right)},\cdots ,\sigma_{P}^{\left(h\right)}\right)}^{⊤},$$
*where $\sigma_{\ell }^{\left(h\right)}$ is the circular standard deviation of the angles $d_{\max(\ell -h,1)}^{\left(h\right)},\cdots ,d_{\ell }^{\left(h\right)}$, which reads [44]:*
$$\sigma_{\ell }^{\left(h\right)}=\sqrt{-2\ln\left|\frac{1}{\min(\ell ,h+1)}\sum_{m=\max(\ell -h,1)}^{\ell }\left(\cos d_{m}^{\left(h\right)}+i\sin d_{m}^{\left(h\right)}\right)\right|}.$$

Consistently, if h=0, the LSD-QTDS is a time series full of zeroes. Figure 7 shows an example of LSD-QDTS computed from a uQTS.

We will work only with these two novel real-valued time series in the study. Specifically, they will be used as the only two predictors in all machine learning algorithms that will be compared to perform the joint segmentation and classification step of the model outlined in Figure 1. This choice is motivated by the fact that these two real-valued time series allow us to visually separate the different types of activity pretty well.

### 2.3. Supervised Classification Models

The pipeline summarized in Figure 1 requires a machine learning algorithm for performing the step of jointly segmenting the signal and classifying the time points into walking or non-walking activities. As part of the experimental design, each time point of the collected data was carefully labelled as being part of a walking or non-walking activity. In addition, we designed a feature space composed of two variables that are the values at each time point of the LM-QDTS and the LSD-QDTS, properly normalized (see Figure 8). We therefore aim to train an appropriate classifier that uses this feature space to predict time point labels into walking or non-walking activities. We used the tidymodels (https://www.tidymodels.org (accessed on 17 March 2022)) consistent ecosystem of R packages to train, tune, test and compare models. We focused on the classification methods that have been so far used in the HAR community. As such, we compared a total of four classification methods, namely:**Decision Tree.** The model is a tree made from a list of *if/then* statements based on thresholding covariates of the feature space. Decision trees rely on a total of three possible hyper-parameters: (i) the tree depth, (ii) the minimum number of time points in a node that grants it authorization to be split further and (iii) the complexity parameter for penalizing complex tree structures to avoid over-fitting. In the tidymodels (https://parsnip.tidymodels.org/reference/decision_tree.html (accessed on 17 March 2022)) ecosystem, decision trees can be estimated by three different algorithms, namely the *Classification And Regression Tree* (CART) algorithm [45] of the rpart package, the C5.0 algorithm [46] of the C50 package and an in-house algorithm from Spark (https://spark.apache.org (accessed on 17 March 2022)) in the sparklyr package. We opted for the CART algorithm, which is the most flexible. In particular, it offers control over the size of the tree, which provides more interpretable trees. We tuned all three hyper-parameters.**Support Vector Machine (SVM).** This model divides the feature space in three parts: (i) an area in which non-walking points are more likely, (ii) an area in which walking points are more likely and (iii) a margin that separates the first two areas [47]. In the tidymodels ecosystem, there are three types of SVM models, namely linear, polynomial and radial basis function (RBF) SVMs. Figure 8 suggests that we should aim at a non-linear border and that walking points are surrounded by non-walking points. This type of topology is usually well captured by *radial basis function support vector machines* (https://scikit-learn.org/stable/auto_examples/classification/plot_classifier_comparison.html (accessed on 17 March 2022)), (https://gist.github.com/WittmannF/60680723ed8dd0cb993051a7448f7805 (accessed on 17 March 2022)). Classification-based RBF SVMs rely on a total of two hyper-parameters: (i) a real positive number representing the cost of predicting a sample within or on the wrong side of the margin and (ii) a real positive number which controls the RBF Gaussian kernel standard deviation. In the tidymodels (https://parsnip.tidymodels.org/reference/svm_rbf.html (accessed on 17 March 2022)) ecosystem, RBF SVMs can be estimated by only one algorithm, namely ksvm (https://parsnip.tidymodels.org/reference/details_svm_rbf_kernlab.html (accessed on 17 March 2022)) from the kernlab package [48]. We tuned both hyper-parameters in this model.**k-Nearest-Neighbors (k-NN).** This model uses the *k* most similar time points from the training set to predict membership of new time points from majority voting. Nearest neighbor models rely on a total of three hyper-parameters: (i) the number *k* of neighbors used for predicting group membership of a new time point, (ii) the Minkowski distance power *p* for assessing how far two time points are in the feature space and (iii) a kernel function used to weight distances between time points. In the tidymodels (https://parsnip.tidymodels.org/reference/nearest_neighbor.html (accessed on 17 March 2022)) ecosystem, k-NN models can be estimated by only one algorithm, namely kknn (https://parsnip.tidymodels.org/reference/details_nearest_neighbor_kknn.html (accessed on 17 March 2022)) [49]. We opted for the default *optimal* weighting kernel [50] as all kernel functions gave the same results. We tuned the distance power *p* and the number *k* of neighbors.**Logistic Regression.** The *logit* transform of the probability of being a walking time point is modeled as a linear combination of the variables in the feature space. It is possible to add a penalty when fitting this model to mitigate overfitting. This adds two hyper-parameters: (i) a global weight that defines the overall amount of penalization for complex models and (ii) a mixture parameter that amounts to the proportion of L1 regularization (LASSO) with respect to L2 regularization (Ridge). In the tidymodels (https://parsnip.tidymodels.org/reference/logistic_reg.html (accessed on 17 March 2022)) ecosystem, logistic regression models can be estimated by six different algorithms. We opted for the simplest one, carried out by the glm (https://parsnip.tidymodels.org/reference/details_logistic_reg_glm.html (accessed on 17 March 2022)) function of the stats package [51]. In effect, all the other engines aim to include the penalty hyper-parameters which we do not need since the feature space is only two-dimensional. Logistic regression models predict class probabilities. We therefore tune a single hyper-parameter which is a threshold that turns such probability into a hard membership.

Table 3 summarizes for each model the hyper-parameters on which they rely and that need to be set along with the tuning grid that was used for each of them. The model tuning will be detailed in the dedicated Section 3.2.

These classification algorithms predict whether a time point corresponds to a walking or non-walking activity independently for each time point. The natural time dependency inherited from the time series structure of the collected data is in effect not accounted for. In the next section, we propose two post-processing strategies to smooth the predictions a posteriori using the intrinsic time dependency.

### 2.4. *A Posteriori* Smoothing

One can use one of the above properly tuned supervised classification method to make a prediction as to whether a time point refers to a walking activity or not. We will refer to these predictions as the **raw predictions**. It is possible to improve upon the raw predictions by acknowledging the intrinsic time dependency in the data collected for a single individual. Specifically, we propose the following ***a posteriori***
**smoothing**:Perform a change point detection in which a change point is defined as a time point with a given type of activity immediately followed by a time point associated with a different activity.Compute the elapsed time between two consecutive change points, starting with the elapsed time until the first detected change point and ending with the elapsed time from the last detected change point.Update the list of change points, keeping only the first out of any two consecutive change points that occurred in less than τ seconds.Label all time points between two change points as referring to walking activities if the number of raw predictions in favor of walking activities in that interval exceeds a given threshold η. Otherwise label them as referring to non-walking activities.

The *a posteriori* smoothing step is applied to each time series and generates **smoothed predictions**. The hyper-parameters τ and η were also tuned using grids of 0:0.05:3 seconds and 0.05:0.05:0.95, respectively. The upper bound on the grid for τ follows from the experimental design in which we included 3-second breaks in between activity changes. Hence, two consecutive change points cannot be considered too close if they are over 3-seconds apart.

## 3. Tuning and Comparing Walking Activity Recognition Models

### 3.1. Data Splitting Scheme

The original data included 28 trials for a total of 240,340 time points. We performed an initial split to obtain:A *training set* with 75% of the trials (i.e., 21 trials for a total of 173,120 time points) that will be used for training (i.e., estimating and tuning) the supervised classification models;A *test set* with 25% of the trials (i.e., seven trials for a total of 67,220 time points) that will be used for computing performance metrics for each tuned model to pick the one that best fits our purpose.

The training set fits two purposes: model estimation and model tuning. The latter refers to optimizing the hyper-parameters of each model to achieve optimal performance, in which *optimal* is to be defined in Section 3.2. We achieve this by resorting to a 5-fold cross-validation scheme. This effectively generates 5 different splits of the training set into:an *analysis set*, with 80% of the trials from the training set (i.e., around 17 trials), used for model estimation;an *assessment set*, with 20% of the trials from the training set (i.e., around four trials), used for evaluating model performance.

This cross-validation strategy provides five independent estimates of each performance metric from which we could obtain a mean value and a standard error estimate. Figure 9 summarizes the full data splitting scheme.

### 3.2. Tuning Strategy

We performed tuning in a two-step fashion: (i) tuning of the supervised classification models and (ii) tuning of the *a posteriori* smoothing using the tuned models. We adopted a common tuning strategy for both steps. When assigning a time point to a given type of activity, we might get it right but we might also make a mistake, as summarized by the following two-way table:

Using the notations from Table 4, since the ultimate goal is to monitor walking activities, what really matters then are the following two aspects:We want to be as sure as possible that a time point assigned to a walking activity (TP + FP) does actually correspond to a walking activity (TP); this is achieved by maximizing the **precision** TP/(TP + FP).We want to make sure that the walking activity recognition pipeline assigns time points to a walking activity a reasonable amount of times; this is achieved by maintaining the **detection prevalence** (TP + FP)/(TP + FP + TN + FN) as close as possible to the actual prevalence of walking activities in the collected data.

Following these two principles, we adopted the following strategy for tuning:Find the combination of hyper-parameters that led to the **detection prevalence closest to the actual prevalence of walking activities**. We will refer to this detection prevalence value as the optimal detection prevalence in the subsequent step.Keep all combinations of hyper-parameters that led to a detection prevalence that stays within one standard error of this optimal detection prevalence.Find the combination of hyper-parameters that led to the **highest precision**. We will refer to this precision value as the optimal precision in the subsequent step.Keep all combinations of hyper-parameters that led to a precision that stays within one standard error of this optimal precision.If there is still more than one remaining combination of hyper-parameters, apply a similar filter for maximizing the accuracy (TP + TN)/(TP + FP + TN + FN).If there is still more than one remaining combination of hyper-parameters, choose the hyper-parameters that yield the **simplest model** according to the following rules:for decision tree: smallest cost_complexity (this penalty avoids over-fitting but the feature space is two-dimensional so it can be kept small) and smallest tree_depth (for favoring interpretability);for radial basis function SVM: smallest cost (smoother margin);for nearest neighbors: smallest neighbors (more memory-effective);for logistic regression: highest threshold (more caution when predicting walking activity);for a posteriori smoothing: first highest η (more caution when predicting walking activity), then smallest distance threshold τ between change points (least possible changes w.r.t. raw predictions).

### 3.3. Performance Metrics for Choosing the Best Walking Activity Recognition Model

We used the tuning strategy described in Section 3.2 to obtain a total of four different walking activity recognition models (WARMs), one for each supervised classification model in Section 2.3. We can then use the test set generated during the data splitting step described in Section 3.1 to compute a number of performance metrics for choosing the best tuned WARM. In this section, we describe these performance metrics, which can be divided into three categories: (i) **classification metrics**, which focus on whether each single time point has been assigned the correct type of activity (Section 3.3.1), (ii) **segmentation metrics**, which focus on whether each activity *session* (set of consecutive time points with common activity) has been assigned the correct type of activity (Section 3.3.2) and (iii) **computation time** (Section 3.3.3).

A common consideration to both classification and segmentation metrics that should be take into account is that the initial manual labelling of the change points (true classes) has been performed within a margin of error of 25 ms. Therefore, prior to computing the performance metrics, we filtered out all manually labelled change points as well the 12 time points before and after each of them.

#### 3.3.1. Classification Metrics

As explained in Section 3.2, the end-goal is to monitor walking activities which led us to tune the WARMs to achieve optimal detection prevalence and precision, and possibly accuracy and area under the ROC curve. We will therefore also compute these metrics by referring to Table 4 applied to the test set.

#### 3.3.2. Segmentation Metrics

The classification metrics focus on assessing performance for well assigning a type of activity to a given time point. Here, we turn the focus to assessing whether each activity *session* (set of consecutive time points with common activity) has been assigned the correct type of activity.

Each measured time series is filled with walking sessions and non-walking sessions. These are deduced by the *true* classes of each time point that have been annotated as part of the experimental design. Figure 10a shows an example of such a segmentation. After running a WARM, we can segment the time series according to the *predicted* classes instead of the true classes, which leads to segments corresponding either to *presumably* walking sessions or to *presumably* non-walking sessions. Figure 10b shows an example of such a segmentation.

Combining both segmentations (the true one and the predicted one), we can achieve a four-class segmentation as shown in Figure 10c,d into true positive (TP) segments (blue), true negative (TN) segments (yellow), false positive (FP) segments (green) and false negative (FN) segments (red). This effectively transposes the confusion matrix from one focused on time points to one focused on activity sessions. From this segmentation-focused confusion matrix, we will compute both **precision** and **accuracy**.

#### 3.3.3. Computation Time

Finally, the computing efficiency of the different machine learning approach tested in the walk detection pipeline is compared using computing time. Indeed, for two machine learning algorithms which have a similar classification and segmentation performance, we prefer the one which is faster than the other one because it would be more implementable in the sensors for a real time walk phase detection.

### 3.4. Results

All the statistical analyses were performed on an iMac with an Apple M1 chip using the R programming language for statistical computing [30].

#### 3.4.1. Tuning of Model Hyper-Parameters

We first tuned the hyper-parameters of each model independently using the strategy outlined in Section 3 and the grids defined in Table 3. The optimal parameters are summarised in Table 5.

The results substantially reveal that (i) the optimal decision tree model has a small depth and no penalty should be used, which was expected given the small dimension of the feature space, (ii) the optimal SVM model favors a rather complex margin surface in order to keep the detection prevalence at the level of the prevalence of walking activities in the data, (iii) the optimal k-NN model uses only three neighbors in order to maintain the detection prevalence at the level of the prevalence of walking activities in the data, and (iv) the logistic regression soft prediction should be thresholded at 0.5, which means that it is not necessary to penalize more abruptly walking activity predictions.

The results also clearly demonstrate that the decision tree outperforms all the other considered models in terms of precision and accuracy. This makes sense because one makes the most out of more complex models such as SVM or k-NN when the feature space is high-dimensional, which is not the case in the present situation. The optimal decision tree is easy to interpret and to visualise as shown by Figure 11. We can observe that, at the root node, the probability for a time point to be associated with a walking activity is of 0.46, which indeed matches the prevalence of walking activities in the data.

#### 3.4.2. Tuning of Smoothing Hyper-Parameters

The tuned models make a prediction independently for each time point. This does not account for the time dependency. In Section 2.4, we exposed an a posteriori smoothing strategy that uses time dependency to adjust the raw predictions. This step depends on two hyper-parameters, namely the minimal distance τ between two consecutive change points and the proportion η of walking time points above which a segment is labeled as walking activity. Table 6 summarizes the optimal values for these parameters, along with the achieved precision and accuracy.

We can observe that, for all four models, the a posteriori smoothing strategy systematically improves upon the raw predictions in terms of precision and accuracy. After tuning the smoothing hyper-parameters, two models appear to equally outperform the others, namely the decision tree and 3-NN models. Hence, the final best tuned WARM consists of using either the decision tree displayed in Figure 11 or the 3-NN model, and of adjusting their raw predictions using the smoothing strategy outlined in Section 2.4 with hyper-parameter values as reported in Table 6.

#### 3.4.3. Performances of Best Tuned WARMs on the Test Set

We finally evaluated all four optimally-tuned WARMs on the test set that we left apart before any tuning steps. Table 7 reports both classification and segmentation metrics (precision and accuracy) as described in Section 3.3.1 and Section 3.3.2, as well as the computation time required to achieve the predictions on the test set.

Overall, all classification metrics (expect for the accuracy of the logistic model) are lower on the test set with regard to the training set, which was, of course, expected. From the perspective of these metrics, the 3-NN model stands out and the decision tree comes in second position. When we balance this with the computation time, the recommended WARM to implement on the sensor chip is clearly the one with the decision tree model.

From the perspective of the segmentation metrics, all WARMs behave similarly. What is interesting from these results is that segmentation metrics are largely lower than classification metrics. This substantially indicates that mis-classification mainly occurs during short periods of time. This is encouraging as it suggests that long walking activities are usually well predicted by the proposed pipelines.

## 4. Discussion and Conclusions

This article presents a novel walking activity recognition model (WARM) for separating walking activity phases from non-walking activity phases, as the data are collected by a wearable sensor for a usage in real life conditions. We trained, tuned and compared a number of possible WARMs on the basis of data collected from a motion sensor with low computing and memory resources that measures the rotation of the hip over time in the form of a time series of unit quaternions. The choice of this experimental design was motivated by the fact that this type of wearable sensors are the most challenging, because one needs to think carefully about a feature space that should be quick to compute and not too heavy to store (since it has to be done on the fly by a sensor with low computing and memory resources), while guaranteeing good classification performances. The contributions of the paper are two-fold: (i) we carefully designed a minimal feature space converting unit quaternion time series into suitable real-valued time series and (ii) we proposed a smoothing step that accounts for the time dependency to improve upon the raw predictions computed by state-of-the-art classification models. The designed feature space along with the choice of a classification model and the a posteriori smoothing step define a WARM. We compared a total of four WARMs with four different classification models (decision tree, radial basis function SVM, k-NN and logistic regression). This was achieved by first carefully designing an experiment that gave us access to the ground truth activity types (walking vs. non-walking time points). We divided the collected data into a training set and a test set that was left apart for a final comparison. The training set was used in a five-fold cross-validation scheme to tune the hyper-parameters of the models and of the smoothing step. The WARMs were compared on the basis of achieved precision and accuracy and computation time. Given the results detailed in Section 3.4, and the constraint to implement the WARM on the sensor chip itself, we recommend to use the WARM with the decision tree model. This WARM achieved a precision of 88% on the training set and of 77% on the test set *while maintaining the detection prevalence at the level of the prevalence of walking activities in the data*, which is quite remarkable, given the simplicity of the feature space.

Since the experimental design targeted the most challenging wearable sensors, the proposed WARM may work seamlessly with any other wearable sensor that collects rotation data of body parts with performances expected to be even better (if, for instance, one wears it closer to the feet). The reported precision and accuracy are to be thought of as lower bounds. When using more powerful wearable sensors, one could also think of enriching the feature space, which might further improve the performance of the WARM. In the future, it could be interesting to add other comparison metrics especially designed for classification problems involving time series, such as the ones proposed by Gensler and Sick [52]. Additionally, we are currently in the process of significantly increasing the database. We will therefore update all the models once it is finalized at a later point in time. In addition, we plan to investigate other transformations of the unit quaternion time series that could complement well the current feature space. The k-NN model also deserves particular attention because it stands out on the test set and could be straightforwardly generalized to directly use the original unit quaternion time series without transformation into real-valued ones. Finally, there are a number of other strategies that we have in mind to account for time dependency that we shall compare.

## Figures and Tables

**Figure 1 sensors-22-03555-f001:**
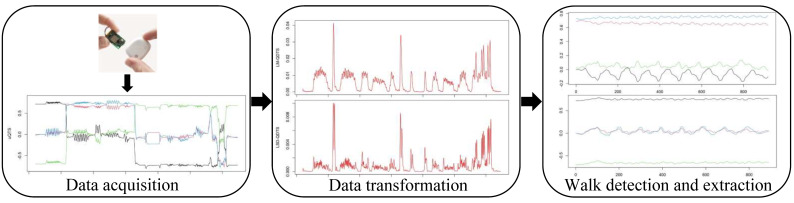
Schematic view of the proposed pipeline. Firstly, by IMU sensors based on the right hip, we collect daily recording of an individual in the form of unit quaternion time series. We then transform them into real-valued unidimensional time series. Finally, we detect and extract the different walk phases of the daily recording by using supervised machine learning algorithms associated with additional post-treatment.

**Figure 2 sensors-22-03555-f002:**
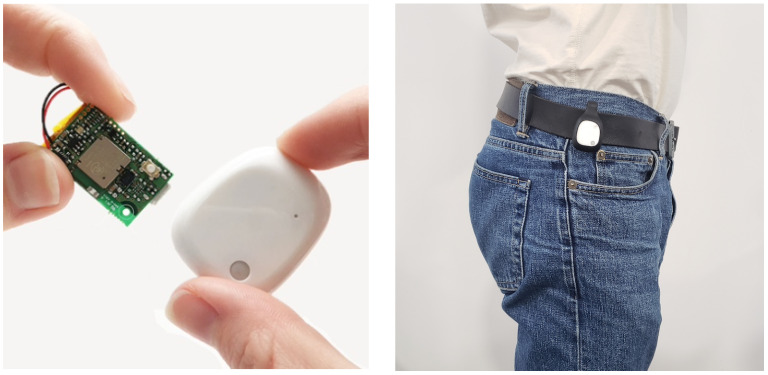
The MetaMotionR sensor device alone (**left panel**) and ready for measurements on an individual (**right panel**).

**Figure 3 sensors-22-03555-f003:**
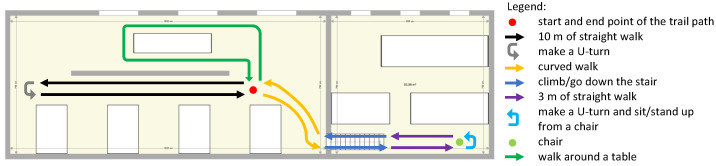
Schematic representation of the path that all subjects followed as part of the experimental design. The total length was approximately 70 m.

**Figure 4 sensors-22-03555-f004:**
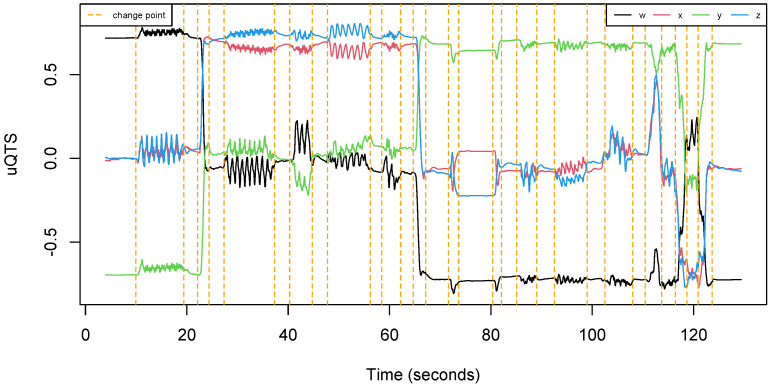
Example of a unit quaternion time series (uQTS) recorded by the MetaMotionR sensor. It traces the hip rotation of an individual while (s)he walked the path described in Figure 3. Orange vertical dotted lines represent the time points at which the type of walking activity changes.

**Figure 5 sensors-22-03555-f005:**
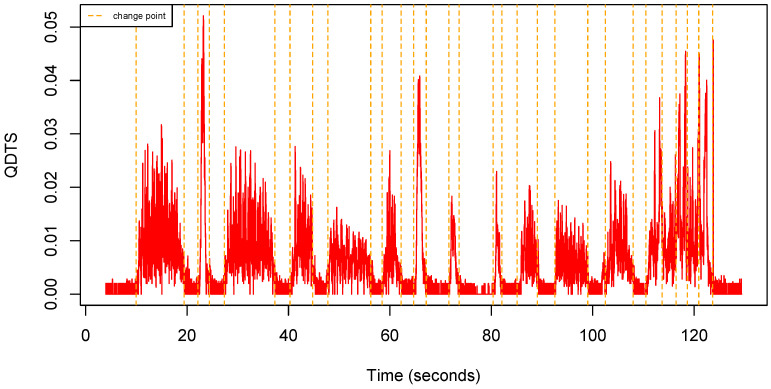
Quaternion distance time series computed from the uQTS shown in Figure 4.

**Figure 6 sensors-22-03555-f006:**
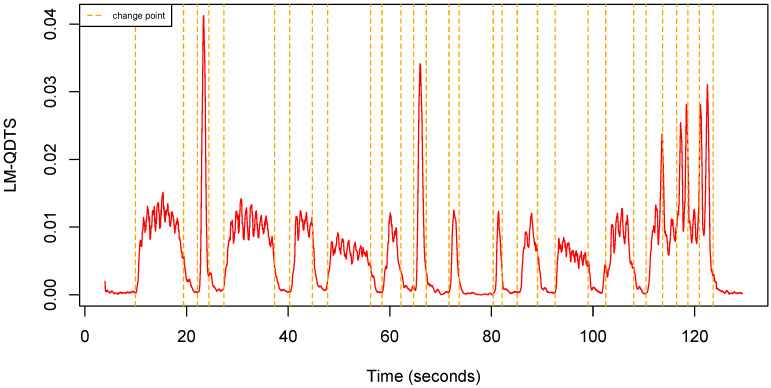
Local Mean QDTS computed from the uQTS in Figure 4 using a sliding window h=50 (0.5 s).

**Figure 7 sensors-22-03555-f007:**
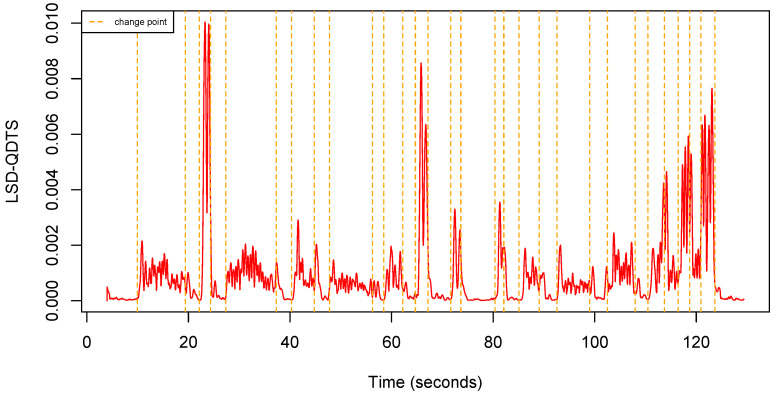
Local standard deviation QDTS computed from the uQTS in Figure 4 using a sliding window h=50 (0.5 s).

**Figure 8 sensors-22-03555-f008:**
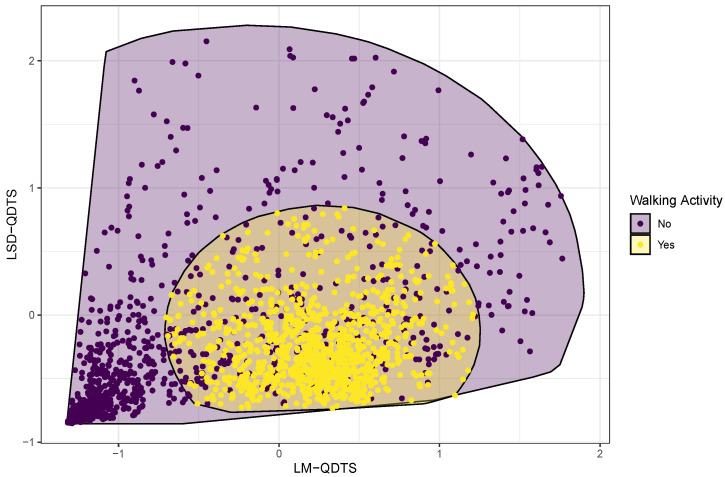
Normalized feature space. Time points referring to walking activities are contrasted with time points referring to non-walking activities using colored points and filled convex hulls.

**Figure 9 sensors-22-03555-f009:**
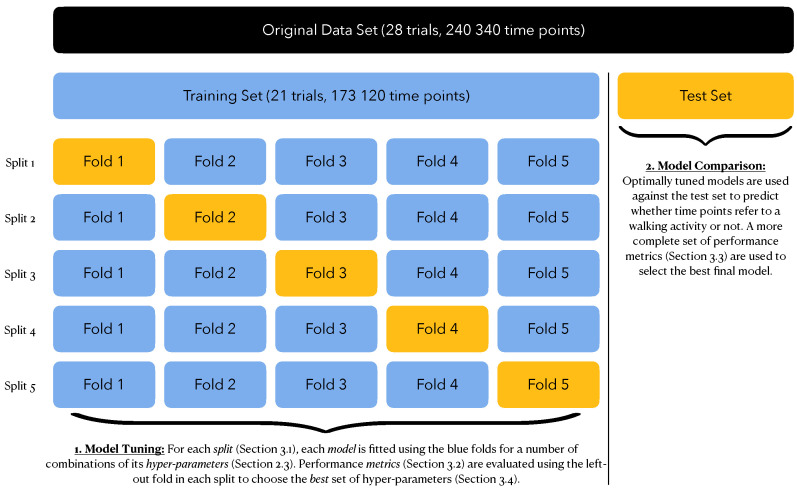
Data Splitting Scheme. The full data set was first divided into a training set (75%) and a test set (25%). The training set was further divided into an analysis set (80%) and an assessment set (20%). A total of five such splits were achieved through five-fold cross-validation.

**Figure 10 sensors-22-03555-f010:**
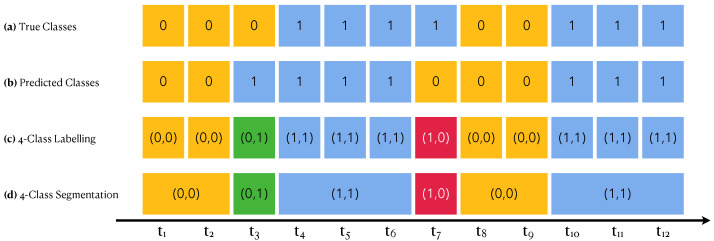
Segmentation Performance. Each cell is a time point. Yellow cells correspond to non-walking activities (encoded by 0); blue ones to walking activities (encoded by 1). Green cells are predicted to be walking sessions but are not, and red ones are the contrary. In a pair code, the left number encodes the true class while the right number encodes the predicted one.

**Figure 11 sensors-22-03555-f011:**
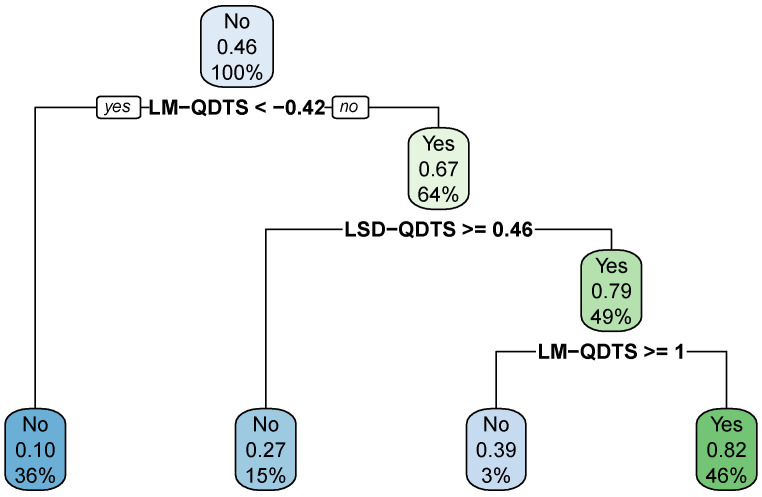
Optimal Decision Tree. Final decision tree estimated on the normalized feature space via the CART algorithm, tuned to maintain the detection prevalence at the level of the prevalence of walking activities in the data and to maximize precision first and accuracy if possible. In each node of the tree, the first row indicates the hard prediction where “No” stands for non-walking activity while “Yes” stands for walking activity; the second row reports the soft prediction, i.e., the probability for a time point to be associated with a walking activity; the third row displays the proportion of data points in the training set that fall into the node.

**Table 1 sensors-22-03555-t001:** Sensor specifications (https://mbientlab.com/tutorials/SensorFusion.html (accessed on 17 March 2022)).

Sensor	Range	Resolution	Sample Rate
Accelerometer	±16 g	16 bit	100 Hz
Gyroscope	$\pm 2000^{\circ }$/s	16 bit	100 Hz
Magnetometer	±1300μT (x,y-axis), ±2500 μT (z-axis)	0.3 μT	25 Hz

**Table 2 sensors-22-03555-t002:** Summary table of the data acquired following the experimental setup.

Volunteer	Number of Trials	Total Recording Time	Age
		(minute)	(year)
V1	8	11	40
V2	9	14	35
V3	11	15	23

**Table 3 sensors-22-03555-t003:** Supervised Classification Models. Summary table of the compared models to perform the joint segmentation and classification of time points into walking and non-walking activities. For each model, hyper-parameters to be tuned are listed (following the tidymodels (https://parsnip.tidymodels.org (accessed on 17 March 2022)) naming conventions) along with the grid that was used for tuning. The notation x:y:z is a shorthand for all values between x and z included with a step of y.

Model	Hyper-Parameters	Tuning Grid
Decision Tree	cost_complexity tree_depth	$10^{-10:1:-1}$ 1:1:10
Radial Basis Function SVM	cost rbf_sigma	$2^{-5:1:5}$ $10^{-10:1:0}$
k-NN	neighbors dist_power	1:2:67 1,2
Logistic Regression	threshold	0.05:0.05:0.95

**Table 4 sensors-22-03555-t004:** Confusion Matrix of Classification Predictions.

	Time point *corresponds to* walking activities	Time point *corresponds to* non-walking activities
Time point *is predicted as* walking activities	True Positive (TP)	False Positive (FP)
Time point *is predicted as* non-walking activities	False Negative (FN)	True Negative (TN)

**Table 5 sensors-22-03555-t005:** Optimal hyper-parameters after model tuning. Summary table of the optimal hyper-parameters obtained using the strategy outlined in Section 3 and the grids defined in Table 3.

Model	Hyper-Parameters	Optimal Value	Precision	Accuracy
Decision Tree	cost_complexity tree_depth	$10^{-10}$ 3	0.81	0.82
Radial Basis Function SVM	cost rbf_sigma	16 $10^{-6}$	0.71	0.72
k-NN	neighbors dist_power	3 2	0.74	0.75
Logistic Regression	threshold	0.5	0.75	0.76

**Table 6 sensors-22-03555-t006:** Optimal hyper-parameters after smoothing tuning. Summary table of the optimal hyper-parameters for the a posteriori smoothing step obtained using the strategy outlined in Section 3 and the grids defined in Table 3.

Model	Minimal Distance τ between Consecutive Change Points	Proportion η of Walking Time Points above Which a Segment Is Labeled as Walking Activity	Precision	Accuracy
Decision Tree	2.20 s	30%	0.88	0.84
Radial Basis Function SVM	2.00 s	30%	0.78	0.74
3-NN	1.60 s	45%	0.89	0.84
Logistic Regression	2.45 s	30%	0.83	0.78

**Table 7 sensors-22-03555-t007:** Performances of the optimally tuned WARMs on the test set.

Model	Classification Metrics		Segmentation Metrics		Time
	Precision	Accuracy	Precision	Accuracy	(seconds)
Decision Tree	0.77	0.84	0.53	0.55	0.4
Radial Basis Function SVM	0.70	0.76	0.57	0.53	181.9
3-NN	0.80	0.85	0.49	0.55	35.7
Logistic Regression	0.75	0.81	0.55	0.53	0.3

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
