# Peer review of "A Novel Walking Activity Recognition Model for Rotation Time Series Collected by a Wearable Sensor in a Free-Living Environment"

_sensors, 2022, doi:10.3390/s22093555_

Round 1
Reviewer 1 Report
The article entitled “A novel walking activity recognition model for rotation time series collected by a motion sensor in a free-living environment” shows walking activity recognition model with a decision tree classifier yield. It may be useful for human machine interface. It could be helpful for sensor development in artificial intelligence. Therefor I highly recommended to publish this article in “Sensors”.
Author Response
We thank you for appreciating the proposed research and for recommending its publication.
Reviewer 2 Report
The authors present techniques for walking activity recognition. The presented work is competitive in both novelty and technical depth. The experimental studies can prove the proposed methods, and the paper is well organized and readable. My further concerns are summarized as follows:
- The data was collected by a motion sensor in the form of a unit quaternion time series. Most of studies employs the raw acceleration, gyroscope and magnetometer data. What is the superiority of using the quaternion data?
- MetaMotionR is the data source for the research work, the authors should provide the key parameters.
- The accuracy and precision results are not as competitive compared to the state-of-the-art. Please explain in the paper.
Reviewer 3 Report
I enjoyed reading this paper. It is well presented and easy to follow. However, I have fundamental concerns with it as a research paper:
1) Ideally, a paper should be general, present novel information to advance the field so the results can be used in a wide sense in the future. However, this paper is very specific, it works only on a particular system and the paper is about getting results on gait from that particular system. Therefore, in my mind, this is not a research paper, it is an application note on an application of gait for that particular system.
2) The paper does not cover the fundamental research on measuring gait using wearables from other papers.
3) Because the authors have focussed their work on a particular system, the results are not applicable to the general research field. Indeed, their results are not compared to other research papers.
With regret, because the whole premise of the paper is not a research paper, I cannot recommend this paper.
Round 2
Reviewer 3 Report
Now that this paper has been generalised, it is much better, more useful.
I have a few comments:
1) Figure 1 is presented before it is discussed in the main text. This is bad practice and I do recommend figure 1 is moved to after it is discussed in the main text.
2) The paper states "The MetaMotionR (MMR) sensors are wearable sensors known as Inertial Measurement Unit (IMU).". This is not causal as IMUs came before your sensor. I suggest "The MetaMotionR (MMR) sensor contains an Inertial Measurement Unit (IMU)."
3) Table 1 has a URL in the main text. It is suggested that this should be a reference, or a footnote like has been done elsewhere in the paper.
4) There seems to be a problem with the references, "Anguita D. et al. [34,42,43],", but [34] is listed as ". Ortiz, J.L.R. Human Activity Recognition", and 43 is listed as "Ortiz, J.L.R. Activity Recognition Experiment". Furthermore, "Anguita D. et al." should be "Anguita et al."
5) Text "The pipeline summarized in Fig. 1 requires" should be "The pipeline summarized in Figure 1 requires"
6) The paper seems to have hyperlinks in the main text to other references , again these should be given in the reference list, or in a numbered footnote.
7) Why is there bold text in the discussions chapter?
8) Text "Since the experimental design targeted the most challenging wearable sensors, the proposed WARM will work seamlessly" should be "Since the experimental design targeted the most challenging wearable sensors, the proposed WARM may work seamlessly"
9) Text "such as the ones proposed in [53]" should be “such as the ones proposed by Gensler and Sick [53] "
10) text "The information provided in this section is adapted from [54,55]." should be "The information provided in this section is adapted from Piórek [54] and Graf [55]."
11) remove "https://doi.org/" from references 4, 5, 13, 14, 28
12) reference 34 and 38 seem to go off the page
13) reference 37 seems to be incomplete.
